# Significant Genes Associated with Mortality and Disease Progression in Grade II and III Glioma

**DOI:** 10.3390/biomedicines12040858

**Published:** 2024-04-12

**Authors:** Bo Mi Choi, Jin Hwan Cheong, Je Il Ryu, Yu Deok Won, Kyueng-Whan Min, Myung-Hoon Han

**Affiliations:** 1Department of Neurosurgery, Hanyang University Guri Hospital, Hanyang University College of Medicine, Guri 11923, Gyeonggi-do, Republic of Korea; 6gagang@hanmail.net (B.M.C.); cjh2324@hanyang.ac.kr (J.H.C.); ryujeil@hanyang.ac.kr (J.I.R.); hidma823@hanmail.net (Y.D.W.); 2Department of Pathology, Uijeongbu Eulji Medical Center, Eulji University School of Medicine, Uijeongbu 11759, Gyeonggi-do, Republic of Korea

**Keywords:** glioma, Wnt/β-catenin signaling, TCGA, survival, gene

## Abstract

Background: The Wnt/β-catenin pathway plays a critical role in the tumorigenesis and maintenance of glioma stem cells. This study aimed to evaluate significant genes associated with the Wnt/β-catenin pathway involved in mortality and disease progression in patients with grade II and III glioma, using the Cancer Genome Atlas (TCGA) database. Methods: We obtained clinicopathological information and mRNA expression data from 515 patients with grade II and III gliomas from the TCGA database. We performed a multivariate Cox regression analysis to identify genes independently associated with glioma prognosis. Results: The analysis of 34 genes involved in Wnt/β-catenin signaling demonstrated that four genes (CER1, FRAT1, FSTL1, and RPSA) related to the Wnt/β-catenin pathway were significantly associated with mortality and disease progression in patients with grade II and III glioma. We also identified additional genes related to the four significant genes of the Wnt/β-catenin pathway mentioned above. The higher expression of BMP2, RPL18A, RPL19, and RPS12 is associated with better outcomes in patients with glioma. Conclusions: Using a large-scale open database, we identified significant genes related to the Wnt/β-catenin signaling pathway associated with mortality and disease progression in patients with grade II and III gliomas.

## 1. Introduction

Gliomas are among the most common primary brain tumors originating from glial cells. They are characterized as diffuse infiltrative tumors. These tumors affect the surrounding brain tissue and cause significant morbidity and mortality. Gliomas are classified as grades I, II, III, and IV based on their pathological characteristics, such as mitotic index, necrosis, microvascular proliferation, and endothelial proliferation [1].

Abundant evidence suggests that tumor cells can exhibit stem cell-like properties and that cancer stemness is a fundamentally important characteristic of malignancy [2]. The upregulation of Wnt/β-catenin signaling inhibits cancer cell differentiation and induces cancer stemness [3]. In addition, a low-grade glioma contains stem cells [4]. The Wnt/β-catenin signaling pathway may play a crucial role in the progression of gliomas and the maintenance of glioma stem cells by inhibiting differentiation [5]. Therefore, if Wnt/β-catenin signaling is upregulated in low-grade glioma, it may promote the stemness of glioma cells, which may lead to a poor prognosis. The Wnt/β-catenin pathway is significantly correlated with the overall survival of patients with glioma and might be a novel prognostic marker [6]. The Cancer Genome Atlas (TCGA), a landmark cancer genomics program that began in 2006 as a collaboration between the National Cancer Institute and the National Human Genome Research Institute, has molecularly characterized over 20,000 primary cancers and matched normal samples spanning 33 cancer types. It is the world’s largest publicly accessible genomic database that catalogs major cancer-causing genomic alterations to achieve a comprehensive “atlas” of cancer genomic profiles (https://gdc.cancer.gov/about-data/publications/pancanatlas (accessed on 12 April 2021) and https://www.cbioportal.org/ (accessed on 12 April 2021)) [7]. The TCGA includes information on digital pathological slides, mRNA expression data, clinicopathological information, and DNA methylation and mutation data. Using the TCGA database, we recently reported an association between glioblastoma (a grade IV glioma) and the *DKK3* gene involved in Wnt/β-catenin signaling [8]. However, we wanted to identify the genes involved in Wnt/β-catenin signaling associated with prognosis in grade II and III gliomas.

The study’s primary goal is to identify significant genes involved in the Wnt/β-catenin signaling pathway associated with mortality and disease progression in patients with grade II and III glioma using the TCGA database. The secondary goal of this study was to evaluate the correlations between these significant genes and examine the possible underlying mechanisms of how they influence each other and the prognosis of glioma.

## 2. Materials and Methods

### 2.1. Study Patients

We recently published a study using 525 glioblastoma multiforme (GBM) cases with information on mRNA expression obtained from the TCGA database [8]. After excluding cases with missing values for important variables (only World Health Organization [WHO] grades II and III gliomas were included), 515 patients with lower-grade glioma (LGG) were included in this study. Before the 2021 WHO classification was released, LGGs were referred to as grade II and III astrocytomas and oligodendrogliomas, respectively [9]. The TCGA lower-grade glioma dataset was completed before 2021; therefore, grade II or III gliomas were categorized as lower-grade gliomas. According to the 2021 WHO classification, it is recommended that WHO grades be described in Arabic numerals instead of Roman numerals. However, because the TCGA data followed the old standard WHO glioma grades, the glioma grades were Romanized in this study [9]. From the TCGA database, we were able to obtain clinical information such as the period of disease progression and death, WHO grade of glioma (grade II or III), glioma histological type (astrocytoma, oligodendroglioma, or oligoastrocytoma), presence of isocitrate dehydrogenase 1 (*IDH1*) mutation, Karnofsky Performance Scale Index, radiation treatment, and laterality of the tumor. The raw data for this study are shown in Appendix A. In the Brain Lower Grade Glioma TCGA dataset (530 cases) from the TCGA database site (https://www.cbioportal.org/ (accessed on 12 April 2021)), researchers can find detailed information, including clinical information, mRNA expression data, pathology reports, and pathology slides for each patient using TCGA ID, which are listed in Appendix A.

Informed consent was not required because data were obtained from the publicly available TCGA database.

### 2.2. Gene Sets Related to the Wnt/β-Catenin Pathway

The Molecular Signature Database (MSigDB) of Gene Set Enrichment Analysis (GSEA) is one of the most widely used and comprehensive gene set databases for performing gene set enrichment analysis [10]. The MSigDB version 7.5.1 in the GSEA (version 4.3.2) (https://www.gsea-msigdb.org/ (accessed on 12 April 2021)) was used to investigate the gene sets related to the Wnt/β-catenin pathway (standard name, ST_WNT_BETA_CATENIN_PATHWAY; systematic name, M17761) [8,11]. A total of 34 genes were related to the Wnt/β-catenin pathway: *AKT1*, *AKT2*, *AKT3*, *ANKRD6*, *APC*, *AXIN1*, *AXIN2*, *CBY1*, *CER1*, *CSNK1A1*, *CTNNB1*, *CXXC4*, *DACT1*, *DKK1*, *DKK2*, *DKK3*, *DKK4*, *DVL1*, *FRAT1*, *FSTL1*, *GSK3A*, *GSK3B*, *LRP1*, *MVP*, *NKD1*, *NKD2*, *PIN1*, *PSEN1*, *PTPRA*, *RPSA*, *SENP2*, *SFRP1*, *TSHB*, and *WIF1*. We extracted mRNA expression data for these 34 genes from the TCGA database of 515 LGG cases (Appendix A).

### 2.3. Bioinformatics Analysis

To investigate the additional genes related to the four selected significant genes (*CER1*, *FRAT1*, *FSTL1*, and *RPSA*) related to the Wnt/β-catenin pathway, we performed pathway-based network analysis using the Search Tool for the Retrieval of Interacting Genes/Proteins (STRING) database version 11.5 hosted by the European Molecular Biology Laboratory (EMBL), Heidelberg, Germany (http://www.string-db.org/ (accessed on 9 July 2021)). STRING provides known and predicted protein–protein association data for numerous organisms based on co-expression analysis, signals across genomes, and the automatic text mining of the biomedical literature [12]. We activated all interaction sources, that is, text mining, experiments, databases, co-expression, neighborhood, gene fusion, and co-occurrence in the STRING setting. The minimum required interaction score was set at 0.400, meaning any interaction power between the two proteins below the medium confidence level was excluded from the analysis [8,13]. To reduce the complexity of the analysis and focus on genes with a strong association with the four selected significant genes, the option of the “max number of interactions to show” in the STRING was set to “no more than 5 interactors”.

Additional bioinformatics analyses were performed using THE Cytoscape (version 3.10.1) software developed by the National Resource for Network Biology (NRNB), University of California, San Diego, CA, USA (https://cytoscape.org/ (accessed on 4 March 2024)). To interpret the biological roles and interactions of the selected significant genes, we used ClueGo and CluePedia plug-ins in Cytoscape, which enable functional gene ontology and pathway network analysis, respectively [14]. We analyzed the annotated biological function pathways based on eight significant genes associated with the prognosis of LGG.

### 2.4. Statistical Analysis

Heatmap analysis was performed using the “pheatmap” package of R software (version 4.1.2).

We calculated the overall survival (OS) and progression-free survival (PFS) rates using Kaplan–Meier analysis for all 34 genes related to the Wnt/β-catenin pathway classified by the cohort’s upper and lower median values of gene expression. We first investigated all statistically significant genes for both OS and PFS in the Kaplan–Meier analysis of the 34 genes. We then performed multivariate Cox regression analysis for the significant genes selected to identify whether they were independently associated with OS and PFS in patients with grade II and III gliomas.

Pearson’s correlation coefficients and significance levels (*p*-values) were calculated to evaluate the relationships between the selected significant genes associated with the OS and PFS in patients with grade II and III glioma using “corrplot” package of R software with the clustering technique (R code: corrplot, M; order = “hclust”; p.mat = p_mat; sig.level = 0.01; method = “square”). Box plots were used to visualize the differences in the expression of selected significant genes between grade II and III gliomas.

Statistical significance was set at *p* < 0.05. All statistical analyses were performed using R software version 4.1.2 and SPSS for Windows (version 24.0 (IBM, Chicago, IL, USA).

## 3. Results

### 3.1. Characteristics of the Study Patients

A total of 515 patients with grade II and III gliomas from the TCGA database were included in the study (Table 1).

The mean age of the patients at diagnosis was 42.9 years, and 44.7% were women. Grade II gliomas account for 48.5% of all gliomas, 37.3% of which are astrocytomas. In addition, 57.5% of the patients underwent radiation therapy. Detailed information is provided in Table 1.

### 3.2. Genes of the Wnt/β-Catenin Pathway Associated with OS and PFS in Patients with Glioma

The heatmap showed log2 fold changes in the Wnt/β-catenin pathway-related gene expressions across the grade II and III glioma cohorts (Figure 1A).

The overall gene expression pattern in the heatmap was relatively homogeneous. In addition, we obtained digitized microscopic images of grade II and III glioma pathology slides from the (TCGA) portal (Figure 1B,C). These slides show that grade III astrocytomas have greater cellularity, increased nuclear atypia, and higher mitotic activity than grade II astrocytomas. However, when we classified patients based on the WHO grade, histological type, and presence of *IDH1* mutation, we did not see any obvious differences in the patterns of Wnt/β-catenin pathway-related gene expressions among the classified groups (Appendix A).

Ten genes involved in the Wnt/β-catenin pathway were found to be significantly associated with both OS and PFS in the Kaplan–Meier survival analysis with a log-rank test (*p* < 0.05) (Table 2).

Four of these ten genes were independently associated with OS and PFS in patients with grade II and III gliomas, as shown by multivariate Cox regression analysis. These four significant genes were: (1) Cerberus 1 (*CER1*) (upper median vs. lower median, OS: hazard ratio (HR), 2.56; *p* < 0.001; PFS: HR, 1.80; *p* < 0.001), (2) the FRAT regulator of WNT signaling pathway 1 (*FRAT1*) (upper median vs. lower median, OS: HR, 0.45; *p* < 0.001; PFS: HR, 0.63; *p* < 0.001), (3) follistatin-like 1 (*FSTL1*) (upper median vs. lower median, OS: HR, 1.92; *p* = 0.003; PFS: HR, 1.68; *p* = 0.002), and (4) ribosomal protein SA (*RPSA*) (upper median vs. lower median, OS: HR, 0.56; *p* = 0.005; PFS: HR, 0.51; *p* < 0.001) (Table 2). The detailed multivariate Cox regression analysis results of these four genes (*CER1*, *FRAT1*, *FSTL1*, and *RPSA*) are presented in Table 3.

We additionally conducted network analysis to investigate the interactions and subcellular localizations among four significant genes (*CER1*, *FRAT1*, *FSTL1*, and *RPSA*) involved in the Wnt/β-catenin pathway (Appendix A). Our analysis revealed intricate interactions among these four significant genes, as well as the distribution of their subcellular localization. We also presented the Kaplan–Meier survival curves for the OS and PFS of the study patients based on the upper and lower median groups of expression of these four genes (Figure 1D–G).

When the study patients were classified according to the WHO grade, *CER1* and *FSTL1* showed significantly higher expression, while *FRAT1* showed significantly lower expression in patients with grade III glioma than in patients with grade II glioma (Appendix A). The four selected genes demonstrated a more significant trend for both OS and PFS in grade III gliomas than in grade II gliomas (Appendix A).

### 3.3. Genes Related to the Four Selected Significant Genes Associated with Both OS and PFS in Patients with Glioma

To broaden the scope of the relationship between the Wnt/β-catenin pathway-related gene and the prognosis of glioma, we identified additional genes closely related to the four selected significant genes using the STRING database (Figure 2A).

The results indicated that: (1) *CER1* exhibited close connections with *BMPR2* (interaction score, 0.921), *BMP2* (interaction score, 0.930), *BMP4* (interaction score, 0.946), *ACVR1B* (interaction score, 0.921), and *TDGF1* (interaction score, 0.947); (2) *FRAT1* demonstrated significant interactions with *CTNNB1* (interaction score, 0.953), *DVL1* (interaction score, 0.972), *AXIN1* (interaction score, 0.996), *GSK3B* (interaction score, 0.990), and *APC* (interaction score, 0.928); (3) *FSTL1* was closely connected to *BMP2* (interaction score, 0.953), *TLR2* (interaction score, 0.972), *DIP2A* (interaction score, 0.996), *BMP4* (interaction score, 0.990), and *SPARC* (interaction score, 0.928); and (4) *RPSA* showed a very close connection with *RPL19* (interaction score, 0.999), *RPS12* (interaction score, 0.999), *RPL18A* (interaction score, 0.999), *RPL35* (interaction score, 0.999), and *RPS* (interaction score, 0.999) (Figure 2A). Eighteen genes were found to be closely related to the four selected genes. We also constructed a heat map to show the log2 fold changes in the expression of these 18 genes in patients with grade II and III gliomas (Figure 2B). The multivariate Cox regression analysis showed that among these eighteen genes, there were another four genes that were independently associated with the OS and PFS of patients with grade II and III glioma: (1) bone morphogenetic protein 2 (*BMP2*) (upper median vs. lower median, OS: HR, 0.39; *p* < 0.001; PFS: HR, 0.41; *p* < 0.001), (2) *RPL18A* (upper median vs. lower median, OS: HR, 0.55; *p* = 0.004; PFS: HR, 0.60; *p* = 0.001), (3) *RPL19* (upper median vs. lower median, OS: HR, 0.50; *p* = 0.001; PFS: HR, 0.59; *p* = 0.001), and (4) *RPS12* (upper median vs. lower median, OS: HR, 0.52; *p* = 0.003; PFS: HR, 0.53; *p* < 0.001) (Table 2). Detailed results of the multivariate Cox regression analysis for *BMP2*, *RPL18A*, *RPL19*, and *RPS12* are presented in Table 3. The Kaplan–Meier survival curves for OS and PFS of the study patients based on the upper and lower median groups of gene expression of these additional significant genes are presented in Figure 2C–F.

When the patients were again divided based on the WHO grade, *BMP2* showed significantly lower expression in grade III gliomas than in grade II gliomas (Appendix A). Although *BMP2* was significantly associated with both OS and PFS in grade II and III gliomas, a more significant difference was observed in grade III gliomas than in grade II gliomas (Appendix A). However, the RPs (*RPL18A*, *RPL19*, and *RPS12*) showed a more distinct trend, which was statistically significant for both OS and PFS only in grade III gliomas.

### 3.4. Correlations between the Selected Eight Significant Genes

The expression levels of eight selected significant genes (*CER1*, *FRAT1*, *FSTL1*, *RPSA*, *BMP2*, *RPL18A*, *RPL19*, and *RPS12*) are presented in Figure 2G. The correlation analysis of these eight genes showed that they were clearly divided into two clusters (all correlation coefficients in boxes had *p*-value < 0.001 [*x* in the box indicates a *p*-value ≥ 0.001]) (Figure 2H). The genes in the first cluster (cluster 1: *CER1* and *FSTL1*) showed a significant positive correlation with each other; those in the second cluster, FRAT1, *BMP2*, and the RP family (cluster 2: *RPSA*, *RPS12*, *RPL18A*, and *RPL19*) also showed a significant positive correlation with each other. However, a significant negative correlation was noted between clusters 1 and 2 (Figure 2H). In particular, the negative correlation between *FSTL1* and *BMP2* was the strongest (correlation coefficient = −0.61).

### 3.5. Functional Gene Ontology and Pathway Network Analyses

We used the ClueGO and Cytoscape’s CluePedia plug-ins to investigate the enriched pathways and protein interaction network between eight key genes involved in the Wnt/β-catenin pathway (*CER1*, *FRAT1*, *FSTL1*, *RPSA*, *BMP2*, *RPL18A*, *RPL19*, *RPS12*) significantly associated with mortality and disease progression in patients with glioma. We found four significant GO terms, which are as follows: “peptide transfer from *p*-site tRNA to the A-site tRNA”, “phosphorylation of phosphor-(Ser45, Thr41) beta-catenin at Ser37 by GSK-3”, “ligand trap binds the ligand BMP2, blocking BMP signaling”, and “regulation of the Wnt signaling pathway” among eight significant genes and one β-catenin gene (*CTNNB1*) (Figure 3).

Overall, the protein interaction network between the eight significant genes and the β-catenin gene shown in Figure 3 can be summarized briefly as follows: the *RPSA*, *RPL19*, *RPL18A*, and *RPS12* genes are closely correlated and are associated with peptide bond formation in the nucleus. *FRAT1* is associated with the β-catenin phosphorylation cascade. *RPS12* and *FRAT1* are associated with the positive regulation of the Wnt signaling pathway. *BMP2* is associated with cellular differentiation and organ induction. *BMP2* is associated with both positive and negative Wnt signaling. *FSTL1* and *CER1* are associated with blocking BMP signaling. 

In this study, we observed that the genes associated with the prognosis of LGG patients seemed to be solely focused on the WNT/β-catenin signaling pathway. Considering that this may result in a biased perspective, we expanded our scope to investigate whether these eight noteworthy genes also interacted with genes from other signaling pathways known to be related to glioma prognosis. Figure 4 shows a tightly woven interaction network between the RP family members (*RPSA*, *RPL18A*, *RPL19*, and *RPS12*) and genes potentially associated with the immune mechanisms of LGG, which have been reported to be related to the prognosis of patients with LGG (*CD2*, *SPN*, *IL18*, *PTPRC*, *GZMA*, and *TLR7*) [15].

These findings suggest that the RP family, traditionally associated with Wnt/β-catenin signaling, may have a broader role in influencing the immune landscape within LGG, potentially impacting patient prognosis. In addition, we recently identified 12 independent genes across 10 oncogenic signaling pathways significantly associated with mortality and disease progression in patients with GBM [11]. Therefore, we explored the potential connections between these twelve significant genes from the ten oncogenic pathways and the eight noteworthy genes identified in this study. Figure 5 presents a detailed network analysis revealing close interactions between the *CER1*, *FRAT1*, *FSTL1*, and *BMP2* genes and 12 independent genes identified across 10 oncogenic signaling pathways that are significantly associated with prognosis in GBM patients.

These findings indicate substantial crosstalk between Wnt/β-catenin signaling and various other oncogenic pathways, underscoring the role of Wnt/β-catenin signaling in the broader oncogenic landscape and its potential impact on glioma prognosis.

### 3.6. Functions of the Selected Eight Significant Genes on Mortality and Disease Progression in Glioma

Based on previously published studies and pathway network analyses, we present schematic illustrations of the possible mechanisms through which these eight significant genes affect mortality and disease progression in patients with grade II and III gliomas (Figure 6).

Briefly, *FSTL1* overexpression stimulates the Wnt/β-catenin pathway to induce tumorigenesis and cancer stem cell maintenance [16,17] and inhibits the *BMP2* pathway, which results in the undifferentiation of cancer cells. Therefore, differentiated glioma cells are induced to become undifferentiated, which may adversely affect the prognosis of patients with glioma. In addition, the overexpression of *CER1*, a BMP antagonist, inhibits *BMP2* to induce undifferentiated glioma cells. *FRAT1* acts on the glycogen synthase kinase 3 (GSK3) signaling network of Wnt/β-catenin signaling to activate β-catenin. The activation of specific RPs impedes the p53 inhibition of MDM2, which may lead to cell cycle arrest, apoptosis, and the differentiation of glioma cells (Figure 6) [18].

## 4. Discussion

The expression of four genes (*CER1*, *FRAT1*, *FSTL1*, and *RPSA*) involved in the Wnt/β-catenin pathway was significantly associated with mortality and disease progression in patients with glioma, especially in those with grade III glioma. Higher expressions of *CER1* and *FSTL1* were associated with poor survival and early glioma recurrence. In contrast, the higher expression of *FRAT1* and *RPSA* in gliomas is associated with better survival and delayed tumor recurrence. To expand the scope of our study, we identified additional genes related to the previously mentioned four significant genes of the Wnt/β-catenin pathway. Of these additional genes, the higher expression of four genes (*BMP2*, *RPL18A*, *RPL19*, and *RPS12*) was also significantly associated with better survival and delayed tumor recurrence in patients with glioma, especially those with grade III glioma. Among these eight significant genes, positive correlations were observed between *CER1* and *FSTL1* and between *FRAT1*, *BMP2*, and the *RP* family (*RPSA*, *RPS12*, *RPL18A*, and *RPL19*). Through bioinformatics network analysis, we found that the eight significant genes identified in our study, correlated with the prognosis of LGG and associated with the Wnt signaling pathway, exhibited close connections with genes reported to be involved in potential immune mechanisms influencing LGG prognosis. Additionally, we discovered that these eight significant genes had close connections with genes belonging to different signaling pathways that have been reported to be associated with the prognosis of GBM. Therefore, we believe these eight to be noteworthy, as these genes are not limited to the Wnt/β-catenin signaling pathway but may also serve as potentially crucial key genes that interact with various signaling pathways important in glioma tumorigenesis.

Up to 70% of low-grade gliomas transform into high-grade gliomas within 10 years, and this transformation is associated with changes in several genes and molecular pathways [19,20]. We recently reported genes differentially expressed between grade II or III gliomas and GBM [21]. In this study, we showed that the role of the *DKK3* gene (Wnt/β-catenin pathway) in grade II or III glioma might be altered in grade IV GBM [21]. Therefore, we do not believe that genes associated with GBM prognosis are necessarily associated with the prognosis of grade II or III gliomas. In addition, because most researchers are more interested in GBM than lower-grade gliomas, most research on prognostic gene markers has been conducted in GBM. Therefore, the genes identified in this study that are associated with the prognosis of grade II or III gliomas are meaningful because they could be helpful as biomarkers or therapeutic targets in clinical practice for diagnosing or treating grade II or III gliomas.

Glioma stem cells (GSCs) undergo continuous self-renewal and have potent tumorigenic potential. They differed from their more differentiated progeny in response to treatment [22,23]. According to previous studies, *FSTL1* activates the Wnt/β-catenin pathway, which is involved in tumorigenesis and cancer stem cell maintenance [16,17]. On the contrary, *FSTL1* suppresses the *BMP2* pathway, which induces cancer cell differentiation [24,25,26,27,28]. Our study also supports this observation and showed a significant negative correlation between the expressions of *FSTL1* and *BMP2*. Therefore, *FSTL1* overexpression may induce the differentiation of glioma cells into undifferentiated glioma stem-like cells, resulting in higher mortality and rapid disease progression in patients with glioma [29,30].

BMPs play a paradoxical role in cancer cell proliferation and differentiation [31]. Our pathway network analyses showed that BMPs enhanced or inhibited the Wnt pathway depending on SMAD4 expression [32]. According to previous studies, *BMP2* significantly inhibits tumor cell proliferation and induces cancer cell autophagy and differentiation [27,28,33,34,35]. *BMP2* acts as a potent tumor suppressor in gastric, renal cell, lung, and colorectal cancers, as well as osteosarcoma, inhibiting tumor growth by reducing the gene expression of oncogenic factors and inducing the differentiation of cancer stem cells [36]. BMP/SMAD4 signaling suppresses WNT-driven dedifferentiation and oncogenesis in the differentiated gut epithelium [37]. Our study also showed that *BMP2* overexpression was significantly associated with better survival and delayed disease progression in patients with glioma. Network analysis in Figure 3 shows that *CER1* and *FSTL1* inhibit *BMP2* signaling. Therefore, we speculated that the inhibition of BMP2/SMAD4 signaling, as illustrated in Figure 6, inhibits glioma cell differentiation, leading to glioma cell stemness, which may lead to a poor prognosis in patients with grade II and grade III gliomas.

*CER1* overexpression was associated with higher mortality and disease progression in the study patients. *CER1* is a BMP antagonist [38,39]. Therefore, we believe that similarly to *FSTL1*, *CER1* may also inhibit the activation of the *BMP2* pathway, which would induce undifferentiation of glioma cells and cause poor prognosis in patients with glioma.

*FRAT1* is an important member of the GSK3 signaling network of Wnt/β-catenin signaling. Its overexpression in gliomas upregulates the intracellular accumulation of β-catenin [40]. *FRAT1* may promote the development of gliomas and is associated with various malignancies [40,41]. We do not know the exact mechanism of why the overexpression of *FRAT1*, a Wnt/β-catenin signaling-related oncogene, improves the prognosis in patients with glioma. However, previous studies on glioma and *FRAT1* have mostly focused on glioblastoma, a WHO grade IV glioma, rather than the grade II or III gliomas considered in this study [42,43]. Also, most previous studies have reported that *FRAT1* expression is positively correlated with increasing WHO glioma grade or the expression level of β-catenin [40,41]. To the best of our knowledge, this is the first study to report a possible association between the level of *FRAT1* expression in glioma cells and the survival rate and disease progression among a relatively large number of patients with glioma. A previous study reported that canonical Wnt signaling was not affected by the absence of *FRAT* in mammals [44]. This may mean that *FRAT* is not an essential component of the canonical Wnt pathway in mammals [44]. In addition, it is possible that the function of *FRAT* is not limited to Wnt signaling and can perform other functions independent of its role in the GSK3 signaling network of Wnt signaling [45].

The fundamental function of the RP family is to stabilize small and large ribosomal subunits and perform additional divergent processes of pre-ribosomal particle assembly, including the folding, stabilization, processing, and transport of rRNA [46]. However, RPs also play various extra-ribosomal roles in cell growth, cell division, and cell death [46,47]. RPs play complex roles in cancer. In addition, emerging evidence has shown that the RP family plays a crucial role in mediating p53 signaling in response to ribosomal stress [18]. In the presence of ribosomal stress, RPs bind to mouse double minute 2 (MDM2) and inhibit MDM2-mediated p53 ubiquitination and degradation to stabilize and activate p53 [18,46]. p53, a tumor suppressor, plays a critical role in suppressing tumorigenesis [48]. In our study, the overexpression of RPs, such as *RPSA*, *RPL18A*, *RPL19*, and *RPS12*, was associated with a better prognosis in patients with glioma. Therefore, we believe that the overexpression of a specific RPs caused by ribosomal stress may induce the activation of p53 through the RP–MDM2–p53 pathway and inhibit glioma tumorigenesis [49].

Our results demonstrate that the eight significant genes related to glioma prognosis tend to have greater statistical significance in grade III than in grade II gliomas. Our study also showed that the expression of genes associated with poor prognosis in gliomas, such as *FSTL1* and *CER1*, was significantly higher in grade III than in grade II gliomas. Conversely, the expression of genes related to good prognosis, such as *FRAT1* and *BMP2*, was significantly lower in grade III than in grade II gliomas. However, the WHO glioma grade did not affect the expression levels of RPs. However, the exact underlying mechanism remains unknown. Based on our findings, we hypothesized that the effects of *FSTL1* and *CER1* in grade III gliomas are stronger, whereas those of *FRAT1* and *BMP2* are weaker. Therefore, the prognosis of patients with grade III glioma may deteriorate more rapidly than patients with grade II glioma. In contrast, if the actions of *FRAT1* and *BMP2* become stronger and those of *FSTL1* and *CER1* become weaker in patients with grade III gliomas, the difference in survival and disease progression between these opposing groups may significantly increase.

Our study had some limitations. First, because this study was retrospective in nature and used TCGA database, it is necessary to conduct further prospective studies to validate the results. We have presented all TCGA data used in this study as supplementary data, as this will allow other researchers to check and verify our results. Second, as the effect of the selected genes on gliomas was not verified through experimental analysis, further in vitro and/or in vivo studies are required. Third, because of the nature of TCGA data, some data, including those on *IDH1* mutation status, were missing, which may have affected the statistical analysis results. Furthermore, the lack of information on the patients’ *IDH1* mutation status in the TCGA lower-grade glioma dataset made it difficult to classify patients according to the 2021 CNS WHO classification. Fourth, the nature of the TCGA lower-grade glioma dataset used in our study led us to combine cases of astrocytomas and oligodendrogliomas, which are molecularly and biologically distinct tumors according to the 2021 CNS WHO classification, into one group for analysis [50]. Therefore, there is a potential for significant bias. Fifth, the results of the heatmap analysis comparing grade II and grade III glioma groups alone cannot prove that there is no difference in Wnt/β-catenin pathway-related gene expression.

## 5. Conclusions

This study was the first to identify significant genes related to the Wnt/β-catenin signaling pathway, which are associated with mortality and disease progression in patients with grade II and III glioma, using a large-scale, open database. We also present possible mechanisms to explain our findings based on previous studies. Although our findings must be verified, they may enhance our understanding of the mechanisms underlying glioma pathophysiology and help develop treatments for patients with glioma.

## Figures and Tables

**Figure 1 biomedicines-12-00858-f001:**
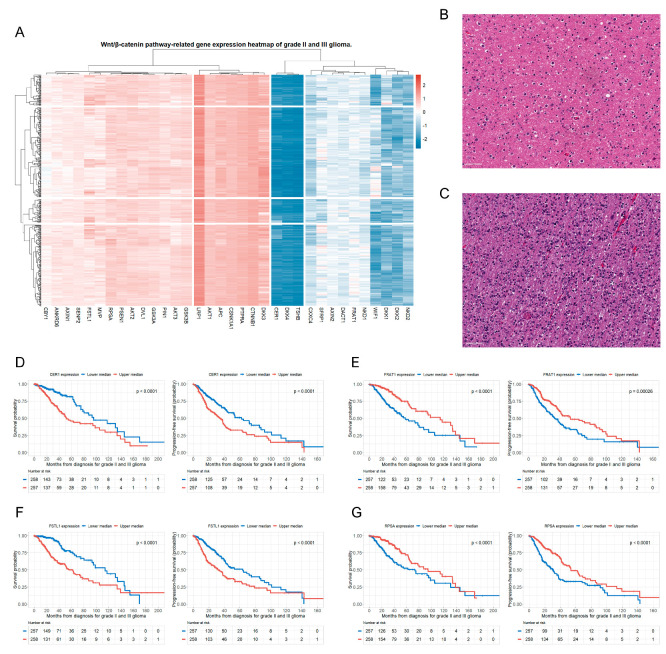
Wnt/β-catenin pathway-related gene expression pattern in patients with grade II and III glioma. Four significant genes related to the Wnt/β-catenin pathway are associated with mortality and disease progression in patients with grade II and III glioma. (**A**) A hierarchically clustered heatmap showing the expression patterns of 34 genes related to the Wnt/β-catenin signaling pathway in patients with grade II and III glioma. Gene expressions are transformed in log2, and color density is displayed, indicating levels of log2 fold changes. Red and blue represent up- and down-regulated expressions in grade II and III glioma, respectively; (**B**) CNS WHO-grade 2 glioma: an infiltrating astrocytoma of low cell density, showing mild nuclear atypia of tumor cells and a dense fibrillar background with mild edema; (**C**) CNS WHO-grade 3 glioma: IDH-mutant astrocytoma showing greater cellularity, nuclear atypia, and increased mitotic activity than that exhibited by WHO-grade 2 astrocytoma; (**D**) OS and PFS rates of patients with glioma based on the upper and lower median groups of *CER1* expression; (**E**) OS and PFS rates of patients with glioma based on the upper and lower median groups of *FRAT1* expression; (**F**) OS and PFS rates of patients with glioma based on the upper and lower median groups of *FSTL1* expression; and (**G**) OS and PFS rates of patients with glioma based on the upper and lower median groups of *RPSA* expression. CNS, central nervous system; WHO, World Health Organization; IDH, isocitrate dehydrogenase; OS, overall survival; PFS, progression-free survival; *CER1*, cerebellar 1; *FRAT1*, FRAT regulator of WNT signaling pathway 1; *FSTL1*, follistatin-like 1; *RPSA*, ribosomal protein SA.

**Figure 2 biomedicines-12-00858-f002:**
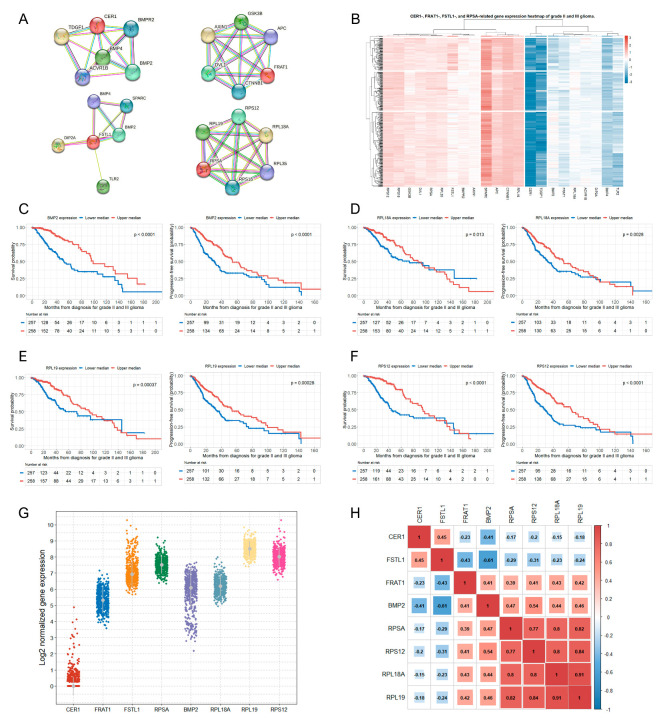
Identification of additional four significant genes associated with mortality and disease progression in patients with grade II and III glioma and correlations between eight significant genes. (**A**) The Wnt/β-catenin pathway-associated protein–protein interaction network of four significant genes (*CER1*, *FRAT1*, *FSTL1*, and *RPSA*) was constructed using a STRING database (V11.5). Five related genes were searched for each significant gene. The thickness of the line between any two proteins represents the degree of confidence in the interaction between the two proteins, with thicker lines indicating higher confidence; (**B**) a hierarchically clustered heatmap showing the expression patterns of the expressions of eighteen genes related to the four significant genes (*CER1*, *FRAT1*, *FSTL1*, and *RPSA*) in patients with grade II and III glioma. Gene expressions are transformed in log2, and color density is displayed, indicating log2 fold changes. Red and blue represent up- and downregulated expressions in grade II and III glioma, respectively; OS and PFS rates of patients with glioma according to the upper and lower median groups of (**C**) *BMP2* expression; (**D**) *RPL18A* expression; (**E**) *RPL19* expression; and (**F**) *RPS12* expression; (**G**) strip plots showing log2-transformed gene mRNA expressions based on the selected eight significant genes; and (**H**) Pearson’s correlation coefficients and significance levels were calculated between the selected eight significant genes. The color-coordinated legend indicates the value and sign of Pearson’s correlation coefficient. The number in the box indicates Pearson’s correlation coefficient. The *x* in the box indicates a *p*-value of ≥0.001. *CER1*, cerebrum 1; *FRAT1*, FRAT regulator of WNT signaling pathway 1; *FSTL1*, follistatin-like 1; *RPSA*, ribosomal protein SA; STRING, Search Tool for the Retrieval of Interacting Genes/Proteins; OS, overall survival; PFS, progression-free survival; *BMP2*, bone morphogenetic protein 2; *RPL18A*, ribosomal protein L18A; *RPL19*, ribosomal protein L19; *RPS12*, ribosomal protein S12.

**Figure 3 biomedicines-12-00858-f003:**
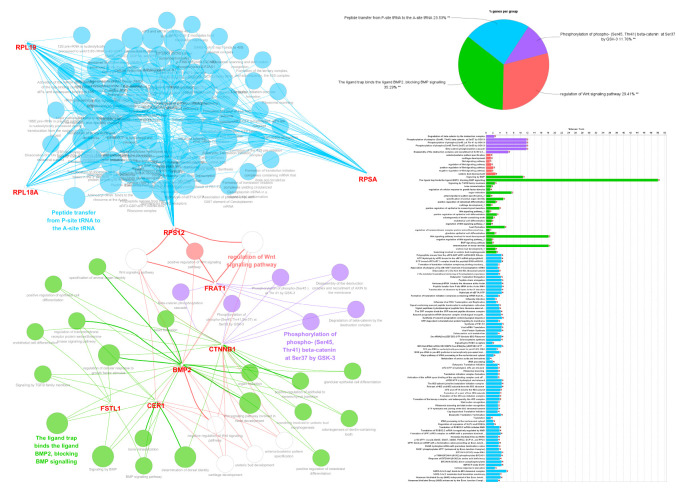
Bioinformatics analysis using Cytoscape with ClueGo and CluePedia plug-ins. The grouping of the networks of significant genes associated with the prognosis of grade II and III gliomas based on functionally enriched GO terms and pathways. GO, gene ontology.

**Figure 4 biomedicines-12-00858-f004:**
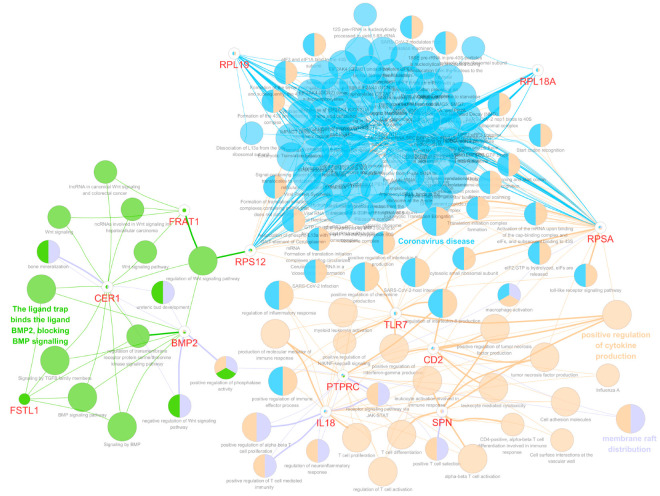
Bioinformatics network analysis visualizing the interactions between eight significant genes identified in this study (*CER1*, *FRAT1*, *FSTL1*, *RPSA*, *BMP2*, *RPL18A*, *RPL19*, and *RPS12*) and genes that are potentially involved in regulating the immune microenvironment and serve as independent prognostic markers for LGG *(CD2*, *SPN*, *IL18*, *PTPRC*, *GZMA*, and *TLR7*). The network was generated using Cytoscape with functional GO terms and biological pathways enrichment. *CER1*, cerberus 1; *FRAT1*, FRAT regulator of WNT signaling pathway 1; *FSTL1*, follistatin-like 1; *RPSA*, ribosomal protein SA; *BMP2*, bone morphogenetic protein 2; *RPL18A*, ribosomal protein L18A; *RPL19*, ribosomal protein L19; *RPS12*, ribosomal protein S12; LGG, lower-grade glioma; *SPN*, sialophorin; *IL*, interleukin; *PTPRC*; protein tyrosine phosphatase receptor type C; *GZMA*, granzyme A; *TLR7*, Toll-like receptor 7; GO, gene ontology.

**Figure 5 biomedicines-12-00858-f005:**
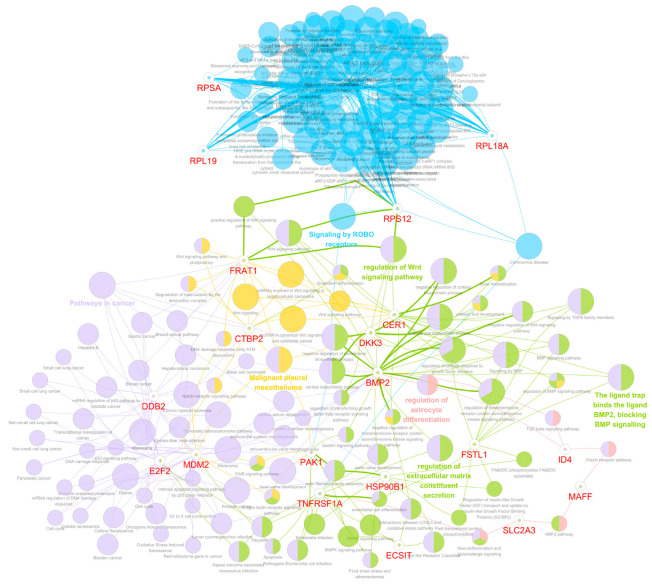
Bioinformatics network analysis visualizing the interactions between eight significant genes discovered in this research (*CER1*, *FRAT1*, *FSTL1*, *RPSA*, *BMP2*, *RPL18A*, *RPL19*, and *RPS12*), and twelve independent genes from ten oncogenic signaling pathways (*E2F2* [cell cycle signaling pathway], *CTBP2* [Notch signaling], *MAFF* [Nrf2 signaling], *SLC2A3* [Nrf2 signaling], *ECSIT* [PI3K signaling], *HSP90B1* [PI3K signaling], *TNFRSF1A* [PI3K signaling], *PAK1* [RTK signaling], *ID4* [TGF-β signaling], *DDB2* [p53 signaling], *MDM2* [p53 and cell cycle signaling], and *DKK3* [Wnt/β-catenin signaling]) that have been significantly associated with prognosis in patients with GBM. The network was generated using Cytoscape with the functional enrichment of GO terms and biological pathways. *CER1*, cerberus 1; *FRAT1*, FRAT regulator of WNT signaling pathway 1; *FSTL1*, follistatin like 1; *RPSA*, ribosomal protein SA; *BMP2*, bone morphogenetic protein 2; *RPL18A*, ribosomal protein L18A; *RPL19*, ribosomal protein L19; *RPS12*, ribosomal protein S12; *E2F2*, E2F transcription factor 2; *CTBP2*, C-terminal-binding protein 2; *MAFF*, MAF bZIP transcription factor F; *Nrf2*, nuclear factor erythroid 2-related factor 2; *SLC2A3*, solute carrier family 2 member 3; *ECSIT*, evolutionarily conserved signaling intermediate in Toll pathways; *PI3K*, phosphatidylinositol 3-kinase; *HSP90B1*, heat shock protein 90 kDa beta member 1; *TNFRSF1A*, tumor necrosis factor receptor superfamily member 1A; *PAK1*, p21 activated kinase 1; *RTK*, receptor tyrosine kinase; *ID4*, inhibitor of DNA binding 4; *TGF-β*, transforming growth factor beta; *DDB2*, damage-specific DNA-binding protein 2; *MDM2*, mouse double minute 2 homolog; *DKK3*, dickkopf-3; GBM, glioblastoma multiforme; GO, gene ontology.

**Figure 6 biomedicines-12-00858-f006:**
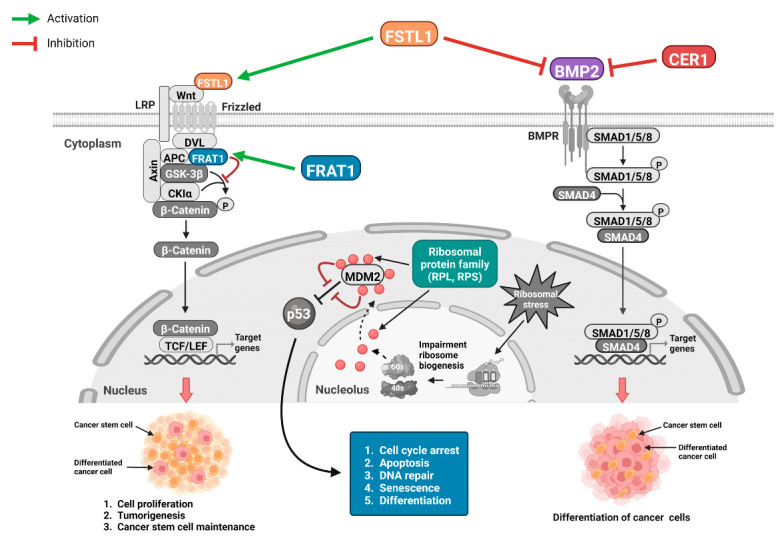
Schematic illustrations of possible roles of the eight significant genes in glioma: Overexpression of FSTL1 activates the Wnt/β-catenin pathway to induce tumorigenesis and cancer stem cell maintenance and inhibits the *BMP2* pathway, which leads to the undifferentiation of cancer cells. Increased *CER1* expression suppresses *BMP2* to induce undifferentiated glioma cells. *FRAT1* is known to act on the *GSK3* signaling network of the Wnt/β-catenin signaling to activate β-catenin. However, based on our findings, the function of *FRAT1* may not be limited to Wnt signaling. It can perform other functions independently of its role in the GSK3 signaling network of Wnt signaling. The overexpression of specific RPs suppresses the p53 inhibition of MDM2, which may lead to glioma cell cycle arrest, apoptosis, and the differentiation of glioma cells. *FSTL1*, follistatin-like 1; *BMP2*, bone morphogenetic protein 2; *CER1*, cerebellar 1; *FRAT1*, FRAT regulator of WNT signaling pathway 1; *GSK3*, glycogen synthase kinase 3; RP, ribosomal protein; MDM, mouse double minute 2 homolog.

**Table 1 biomedicines-12-00858-t001:** Clinical characteristics of patients with grade II and III glioma.

Characteristics	Total
Number	515
Sex, female, n (%)	230 (44.7)
Age at diagnosis of glioma, mean ± SD, y	42.9 ± 13.4
Time duration between glioma diagnosis and death (months), mean ± SD	31.7 ± 31.5
Time duration between glioma diagnosis and disease progression (months), mean ± SD	25.8 ± 25.7
WHO-grade glioma, n (%)	
Grade II	250 (48.5)
Grade III	265 (51.5)
Glioma histological type, n (%)	
Astrocytoma	194 (37.7)
Oligodendroglioma	191 (37.1)
Oligoastrocytoma	130 (25.2)
*IDH1* mutation, n (%)	
Yes	91 (17.7)
No	34 (6.6)
Missing data	390 (75.7)
Karnofsky Performance Scale Index, median (IQR)	80.0 (80.0–90.0)
Missing data, n (%)	52 (10.1)
Radiation treatment, n (%)	
Yes	296 (57.5)
No	185 (35.9)
Missing data	34 (6.6)
Tumor laterality, n (%)	
Left	250 (48.5)
Right	253 (49.1)
Missing data	12 (2.3)

SD, standard deviation; WHO, World Health Organization; *IDH1*, isocitrate dehydrogenase; IQR, interquartile range.

**Table 2 biomedicines-12-00858-t002:** Multivariate Cox analyses of the genes significantly associated with overall survival and progression-free survival in the Kaplan–Meier analysis in patients with grade II and III glioma.

	Overall Survival			Progression-Free Survival		
	Log-Rank Test (Kaplan–Meier Analysis)	Multivariate Cox Regression Analysis *		Log-Rank Test (Kaplan–Meier Analysis)	Multivariate Cox Regression Analysis *	
1. Wnt/β-catenin signaling (34 genes were analyzed) (upper median vs. lower median)	** *p* **	**HR (95% CI)**	** *p* **	** *p* **	**HR (95% CI)**	** *p* **
*AKT2*	<0.001	1.43 (0.93–2.21)	0.106	<0.001	1.54 (1.10–2.18)	0.014
*CER1*	**<0.001**	**2.56 (1.69–3.89)**	**<0.001**	**<0.001**	**1.80 (1.32–2.46)**	**<0.001**
*CXXC4*	0.002	0.71 (0.47–1.06)	0.095	<0.001	0.67 (0.49–0.92)	0.014
*DKK3*	0.035	1.44 (0.97–2.15)	0.075	0.031	1.32 (0.98–1.79)	0.071
*FRAT1*	**<0.001**	**0.45 (0.30–0.68)**	**<0.001**	**<0.001**	**0.63 (0.47–0.86)**	**0.004**
*FSTL1*	**<0.001**	**1.92 (1.24–2.98)**	**0.003**	**<0.001**	**1.68 (1.21–2.33)**	**0.002**
*NKD1*	0.008	0.94 (0.62–1.41)	0.752	0.031	0.90 (0.66–1.24)	0.528
*PTPRA*	0.002	0.86 (0.58–1.29)	0.472	0.001	0.73 (0.54–1.00)	0.051
*RPSA*	**<0.001**	**0.56 (0.37–0.84)**	**0.005**	**<0.001**	**0.51 (0.37–0.69)**	**<0.001**
*SENP2*	0.009	0.67 (0.45–1.01)	0.054	0.022	0.67 (0.49–0.92)	0.013
2. CER1-related genes (5 genes were analyzed) (upper median vs. lower median)						
*ACVR1B*	0.038	0.71 (0.47–1.06)	0.096	0.026	0.79 (0.58–1.08)	0.139
*BMP2*	**<0.001**	**0.39 (0.25–0.62)**	**<0.001**	**<0.001**	**0.41 (0.30–0.58)**	**<0.001**
*BMP4*	0.005	0.66 (0.44–1.00)	0.049	0.029	0.77 (0.56–1.04)	0.091
3. FRAT1-related genes (5 genes were analyzed) (upper median vs. lower median)						
None	…	…	…	…	…	…
4. FSTL1-related genes (5 genes were analyzed) (upper median vs. lower median)						
*BMP2*	**<0.001**	**0.39 (0.25–0.62)**	**<0.001**	**<0.001**	**0.41 (0.30–0.58)**	**<0.001**
*BMP4*	0.005	0.66 (0.44–1.00)	0.049	0.029	0.77 (0.56–1.04)	0.091
*SPARC*	0.001	0.82 (0.53–1.26)	0.360	<0.001	0.71 (0.52–0.98)	0.035
*TLR2*	<0.001	1.60 (0.99–2.56)	0.053	<0.001	1.58 (1.12–2.24)	0.010
5. RPSA-related genes (5 genes were analyzed) (upper median vs. lower median)						
*RPL18A*	**0.013**	**0.55 (0.37–0.83)**	**0.004**	**0.003**	**0.60 (0.44–0.82)**	**0.001**
*RPL19*	**<0.001**	**0.50 (0.33–0.75)**	**0.001**	**<0.001**	**0.59 (0.43–0.80)**	**0.001**
*RPS12*	**<0.001**	**0.52 (0.34–0.80)**	**0.003**	**<0.001**	**0.53 (0.39–0.73)**	**<0.001**

HR, hazard ratio; CI, confidence interval; *AKT2*, AKT serine/threonine kinase 2; *CER1*, Cerberus-1; *CXXC4*, CXXC finger protein 4; *DKK3*, Dickkopf WNT signaling pathway inhibitor 3; *FRAT1*, FRAT regulator of WNT signaling pathway 1; *FSTL1*, follistatin-like 1; *NKD1*, NKD inhibitor of WNT signaling pathway 1; *PTPRA*, protein tyrosine phosphatase receptor type A; *RPSA*, ribosomal protein SA; *SENP2*, SUMO-specific peptidase 2; *BMP2*, bone morphogenetic protein 2; *ACVR1B*, activin A receptor type 1B; *SPARC*, secreted protein acidic and cysteine-rich; *TLR2*, Toll-like receptor 2; *RPL18A*, ribosomal protein L18A; *RPL19*, ribosomal protein L19; *RPS12*, ribosomal protein S12. The rows containing genes showing *p* < 0.05 in both overall survival and progression-free survival of multivariate Cox regression analyses are shown in bold. * Adjusted for sex (female vs. male), age (continuous variable), WHO grade (grade III vs. grade II), histological type (oligodendroglioma and oligoastrocytoma vs. astrocytoma), Karnofsky Performance Scale Index, radiation treatment, and tumor laterality (left vs. right).

**Table 3 biomedicines-12-00858-t003:** Detailed information of multivariate Cox analyses of the selected eight significant genes in patients with grade II and III glioma.

Multivariate Cox Analyses of the Selected Four Significant Genes Related to Wnt/β-Catenin Signaling								
Overall Survival								
	*CER1* *		*FRAT1* *		*FSTL1* *		*RPSA* *	
Variable	HR (95% CI)	*p*	HR (95% CI)	*p*	HR (95% CI)	*p*	HR (95% CI)	*p*
Female (vs. male)	0.81 (0.54–1.20)	0.287	0.82 (0.55–1.22)	0.335	0.93 (0.62–1.38)	0.705	0.84 (0.56–1.26)	0.399
Age	**1.06 (1.05–1.08)**	**<0.001**	**1.06 (1.04–1.08)**	**<0.001**	**1.06 (1.04–1.08)**	**<0.001**	**1.06 (1.04–1.08)**	**<0.001**
WHO grade III (vs. grade II)	**2.16 (1.37–3.41)**	**0.001**	**2.29 (1.44–3.62)**	**<0.001**	**1.94 (1.23–3.04)**	**0.004**	**2.10 (1.34–3.29)**	**0.001**
Histological type								
Oligodendroglioma (vs. astrocytoma)	**0.49 (0.30–0.78)**	**0.003**	**0.53 (0.34–0.85)**	**0.008**	**0.60 (0.37–0.97)**	**0.038**	**0.55 (0.34–0.89)**	**0.014**
Oligoastrocytoma (vs. astrocytoma)	0.68 (0.40–1.15)	0.151	0.67 (0.39–1.13)	0.133	0.64 (0.38–1.07)	0.090	0.66 (0.39–1.11)	0.118
Karnofsky Performance Scale Index	0.99 (0.97–1.01)	0.163	1.00 (0.98–1.01)	0.518	0.99 (0.98–1.01)	0.429	0.99 (0.98–1.01)	0.415
Radiation treatment	1.10 (0.66–1.85)	0.713	1.13 (0.68–1.88)	0.645	1.03 (0.61–1.72)	0.922	1.09 (0.65–1.82)	0.741
Tumor laterality, left (vs. right)	0.99 (0.66–1.48)	0.967	1.14 (0.77–1.70)	0.512	1.21 (0.81–1.81)	0.348	1.20 (0.80–1.78)	0.381
*CER1*, upper median (vs. lower median)	**2.56 (1.69–3.89)**	**<0.001**	N/A		N/A		N/A	
*FRAT1*, upper median (vs. lower median)	N/A		**0.45 (0.30–0.68)**	**<0.001**	N/A		N/A	
*FSTL1*, upper median (vs. lower median)	N/A		N/A		**1.92 (1.24–2.98)**	**0.003**	N/A	
*RPSA*, upper median (vs. lower median)	N/A		N/A		N/A		**0.56 (0.37–0.84)**	**0.005**
Progression-free survival								
	*CER1* *		*FRAT1* *		*FSTL1* *		*RPSA* *	
Variable	HR (95% CI)	*p*	HR (95% CI)	*p*	HR (95% CI)	*p*	HR (95% CI)	*p*
Female (vs. male)	1.01 (0.75–1.37)	0.932	1.07 (0.79–1.44)	0.657	1.13 (0.84–1.53)	0.415	1.01 (0.74–1.36)	0.961
Age	**1.04 (1.03–1.05)**	**<0.001**	**1.03 (1.02–1.05)**	**<0.001**	**1.04 (1.02–1.05)**	**<0.001**	**1.04 (1.02–1.05)**	**<0.001**
WHO grade III (vs. grade II)	1.35 (0.96–1.91)	0.084	1.39 (0.99–1.96)	0.058	1.30 (0.93–1.84)	0.130	1.36 (0.97–1.90)	0.075
Histological type								
Oligodendroglioma (vs. astrocytoma)	**0.52 (0.36–0.75)**	**0.001**	**0.53 (0.37–0.76)**	**0.001**	**0.61 (0.42–0.88)**	**0.009**	**0.53 (0.37–0.76)**	**0.001**
Oligoastrocytoma (vs. astrocytoma)	**0.57 (0.38–0.86)**	**0.007**	**0.57 (0.38–0.86)**	**0.007**	**0.57 (0.38–0.85)**	**0.006**	**0.54 (0.36–0.81)**	**0.003**
Karnofsky Performance Scale Index	0.99 (0.98–1.01)	0.253	1.00 (0.99–1.01)	0.704	1.00 (0.99–1.01)	0.670	1.00 (0.99–1.01)	0.619
Radiation treatment	0.76 (0.53–1.09)	0.139	0.77 (0.54–1.11)	0.163	0.73 (0.51–1.05)	0.088	**0.69 (0.48–0.99)**	**0.044**
Tumor laterality, left (vs. right)	1.12 (0.82–1.51)	0.481	1.19 (0.88–1.62)	0.255	1.20 (0.89–1.63)	0.239	1.19 (0.88–1.61)	0.262
*CER1*, upper median (vs. lower median)	**1.80 (1.32–2.46)**	**<0.001**	N/A		N/A		N/A	
*FRAT1*, upper median (vs. lower median)	N/A		**0.63 (0.47–0.86)**	**0.004**	N/A		N/A	
*FSTL1*, upper median (vs. lower median)	N/A		N/A		**1.68 (1.21–2.33)**	**0.002**	N/A	
*RPSA*, upper median (vs. lower median)	N/A		N/A		N/A		**0.51 (0.37–0.69)**	**<0.001**
Multivariate Cox analyses of the selected four significant genes related to *CER1*, *FRAT1*, *FSTL1*, and *RPSA*								
Overall survival								
	*BMP2* *		*RPL18A* *		*RPL19* *		*RPS12* *	
Variable	HR (95% CI)	*p*	HR (95% CI)	*p*	HR (95% CI)	*p*	HR (95% CI)	*p*
Female (vs. male)	0.89 (0.60–1.32)	0.560	0.87 (0.59–1.29)	0.492	0.86 (0.58–1.28)	0.459	0.85 (0.57–1.26)	0.407
Age	**1.06 (1.04–1.08)**	**<0.001**	**1.06 (1.04–1.08)**	**<0.001**	**1.06 (1.04–1.07)**	**<0.001**	**1.06 (1.04–1.07)**	**<0.001**
WHO grade III (vs. grade II)	**2.04 (1.30–3.20)**	**0.002**	**2.19 (1.39–3.46)**	**0.001**	**2.20 (1.40–3.47)**	**0.001**	**2.10 (1.34–3.31)**	**0.001**
Histological type								
Oligodendroglioma (vs. astrocytoma)	0.73 (0.45–1.19)	0.209	**0.51 (0.32–0.82)**	**0.005**	**0.52 (0.33–0.83)**	**0.006**	**0.64 (0.39–1.03)**	**0.064**
Oligoastrocytoma (vs. astrocytoma)	0.76 (0.45–1.29)	0.306	0.61 (0.36–1.03)	0.065	0.60 (0.36–1.02)	0.061	0.72 (0.42–1.22)	0.223
Karnofsky Performance Scale Index	0.99 (0.98–1.01)	0.358	0.99 (0.98–1.01)	0.465	0.99 (0.98–1.01)	0.429	1.00 (0.98–1.01)	0.540
Radiation treatment	1.08 (0.64–1.80)	0.782	1.16 (0.69–1.94)	0.583	1.22 (0.73–2.06)	0.449	1.08 (0.64–1.82)	0.768
Tumor laterality, left (vs. right)	1.28 (0.86–1.91)	0.228	1.17 (0.78–1.74)	0.445	1.18 (0.80–1.76)	0.407	1.16 (0.78–1.72)	0.472
*BMP2*, upper median (vs. lower median)	**0.39 (0.25–0.62)**	**<0.001**	N/A		N/A		N/A	
*RPL18A*, upper median (vs. lower median)	N/A		**0.55 (0.37–0.83)**	**0.004**	N/A		N/A	
*RPL19*, upper median (vs. lower median)	N/A		N/A		**0.50 (0.33–0.75)**	**0.001**	N/A	
*RPS12*, upper median (vs. lower median)	N/A		N/A		N/A		**0.52 (0.34–0.80)**	**0.003**
Progression-free survival								
	*BMP2* *		*RPL18A* *		*RPL19* *		*RPS12* *	
Variable	HR (95% CI)	*p*	HR (95% CI)	*p*	HR (95% CI)	*p*	HR (95% CI)	*p*
Female (vs. male)	1.05 (0.78–1.41)	0.767	1.02 (0.75–1.38)	0.890	1.03 (0.76–1.39)	0.875	1.00 (0.74–1.36)	1.000
Age	**1.03 (1.02–1.05)**	**<0.001**	**1.04 (1.02–1.05)**	**<0.001**	**1.03 (1.02–1.05)**	**<0.001**	**1.03 (1.02–1.05)**	**<0.001**
WHO grade III (vs. grade II)	1.36 (0.96–1.91)	0.082	1.41 (1.00–1.99)	0.051	1.39 (0.99–1.95)	0.059	1.36 (0.97–1.91)	0.078
Histological type								
Oligodendroglioma (vs. astrocytoma)	0.71 (0.48–1.03)	0.070	**0.51 (0.36–0.74)**	**<0.001**	**0.52 (0.36–0.75)**	**<0.001**	**0.60 (0.42–0.87)**	**0.007**
Oligoastrocytoma (vs. astrocytoma)	**0.61 (0.40–0.92)**	**0.018**	**0.53 (0.35–0.79)**	**0.002**	**0.54 (0.36–0.81)**	**0.003**	**0.58 (0.39–0.88)**	**0.010**
Karnofsky Performance Scale Index	1.00 (0.98–1.01)	0.511	1.00 (0.99–1.01)	0.709	1.00 (0.99–1.01)	0.627	1.00 (0.99–1.01)	0.735
Radiation treatment	0.70 (0.49–1.01)	0.057	0.78 (0.54–1.12)	0.170	0.80 (0.55–1.15)	0.219	0.72 (0.50–1.03)	0.073
Tumor laterality, left (vs. right)	1.20 (0.89–1.62)	0.241	1.19 (0.88–1.61)	0.257	1.19 (0.88–1.61)	0.259	1.17 (0.87–1.58)	0.307
*BMP2*, upper median (vs. lower median)	**0.41 (0.30–0.58)**	**<0.001**	N/A		N/A		N/A	
*RPL18A*, upper median (vs. lower median)	N/A		**0.60 (0.44–0.82)**	**0.001**	N/A		N/A	
*RPL19*, upper median (vs. lower median)	N/A		N/A		**0.59 (0.43–0.80)**	**0.001**	N/A	
*RPS12*, upper median (vs. lower median)	N/A		N/A		N/A		**0.53 (0.39–0.73)**	**<0.001**

*CER1*, cerberus 1; *FRAT1*, FRAT regulator of WNT signaling pathway 1; *FSTL1*, follistatin-like 1; *RPSA*, ribosomal protein SA; *BMP2*, bone morphogenetic protein 2; *RPL18A*, ribosomal protein L18A; *RPL19*, ribosomal protein L19; *RPS12*, ribosomal protein S12; HR, hazard ratio; CI, confidence interval; WHO, World Health Organization; N/A, not available; *p* < 0.05 is shown in bold. * Adjusted for sex (female vs. male), age (continuous variable), WHO grade (grade III vs. grade II), histological type (oligodendroglioma and oligoastrocytoma vs. astrocytoma), Karnofsky Performance Scale Index, radiation treatment, and tumor laterality (left vs. right).

## Data Availability

The raw data supporting the conclusions of this article will be made available by the authors on request.

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
