# Peer review of "Significant Genes Associated with Mortality and Disease Progression in Grade II and III Glioma"

_biomedicines, 2024, doi:10.3390/biomedicines12040858_

Round 1

Reviewer 1 Report

Comments and Suggestions for Authors

The authors attempt to explore and identify the critical role in Wnt/B-catenin pathway associated genes with patients' mortality and disease progression within grade II and grade III gliomas population by using 515 cases reported in TCGA database. While the study is attractive, there is a major concern within the study.

The study focused solely on WNT/B-catenin signalling cascade, which only contain 34 genes in total while ignored underlying factors that may shed more insights in mortality and disease progression. Importantly, Wnt/B-catenin signalling cascade is not the sole or main signalling pathway in glioma tumourigenesis. As a result, the findings loss its novelty and applicability for other glioma population.

Author Response

Significant genes associated with mortality and disease progression in grade II and III glioma

Bo Mi Choi, Jin Hwan Cheong, Je Il Ryu, Yu Deok Won, Kyueng-Whan Min, Myung-Hoon Han

Responses to reviewer comments

Editorial Board

Biomedicines

Dear Reviewers and Editorial Staffs of Biomedicines:

We sincerely thank you for your thorough review of our manuscript. We have considered the comments and suggestions with care and addressed the critical issues highlighted by the reviewers. 

These responses and revisions have resulted from the authors’ hard work and sincerity in addressing the suggestions of the reviewers and editors. We are grateful for your guidance in improving the scientific and literary qualities of our work. During the revision process, we underwent English revisions throughout the entire paper.

This manuscript has not been published or presented elsewhere in part or in its entirety and is not under consideration by another journal. Given the use of anonymized data and the retrospective nature of the study, informed consent was waived. The study design was approved by the appropriate ethics review board. We have read and understood your journal’s policies and believe that neither the manuscript nor the study violates them. The authors declare no conflicts of interest.

Thank you for your consideration. I look forward to hearing from you.

Sincerely,

Myung-Hoon Han, M.D., Ph.D.

Department of Neurosurgery, Hanyang University Guri Hospital, 153 Gyeongchun-ro,

Guri, Gyonggi-do, Republic of Korea

Tel: +82-31-560-2326

Fax: +82.31-560-2327

  Kyueng-Whan Min, M.D., Ph.D.

Department of Pathology Uijeongbu Eulji Medical Center, Eulji University School of Medicine, Uijeongbu, Gyeonggi-do, Republic of Korea

Tel: +82-31-560-2496; Fax: +82-31-560-2339

ORCID ID: 0000-0002-4757-9211

Reviewer #1

The authors attempt to explore and identify the critical role in Wnt/B-catenin pathway associated genes with patients' mortality and disease progression within grade II and grade III gliomas population by using 515 cases reported in TCGA database. While the study is attractive, there is a major concern within the study.

The study focused solely on WNT/B-catenin signalling cascade, which only contain 34 genes in total while ignored underlying factors that may shed more insights in mortality and disease progression. Importantly, Wnt/B-catenin signalling cascade is not the sole or main signalling pathway in glioma tumourigenesis. As a result, the findings loss its novelty and applicability for other glioma population.

Thank you  for your valuable comment. As noted in the introduction, our initial selection of the Wnt/β-catenin signaling pathway was due to the known association of its upregulation with inhibiting cancer cell differentiation and inducing cancer stemness (Fodde R, et al. Wnt/beta-catenin signaling in cancer stemness and malignant behavior. Curr Opin Cell Biol 2007;19:150–8). Furthermore, this pathway is believed to play a pivotal role in the progression of gliomas and the maintenance of glioma stem cells by preventing differentiation (Denysenko T, et al. WNT/β-catenin Signaling Pathway and Downstream Modulators in Low- and High-grade Glioma. Cancer Genomics & Proteomics 2016;13:31–45). Therefore, we hypothesized that if Wnt/β-catenin signaling is upregulated in low-grade glioma, it could promote glioma cell stemness, potentially leading to a poorer prognosis. This hypothesis is supported by findings that the Wnt/β-catenin pathway is significantly correlated with the overall survival of patients with glioma and may serve as a novel prognostic marker (Liu C, et al. Wnt/beta-Catenin pathway in human glioma: expression pattern and clinical/prognostic correlations. Clin Exp Med 2011;11:105–12).

In agreement with the reviewer’s comments and inspired by the insights provided, we conducted additional bioinformatics network analyses to assess whether the eight significant genes identified in our research have potential associations with genes related to the immune mechanisms of LGG, which have been implicated in the prognosis of LGG patients (He Y, et al. Identification of prognosis-related gene features in low-grade glioma based on ssGSEA. Front Oncol 2022;12:1056623). Moreover, we explored the interactions between these eight significant genes and 12 independent genes from 10 oncogenic signaling pathways, recently reported to be significantly associated with mortality and disease progression in patients with GBM, through further bioinformatics network analysis using Cytoscape.

Through additional bioinformatics network analysis, we have intriguingly found that the eight significant genes identified in our research, which are correlated with the prognosis of LGG and are associated with the Wnt signaling pathway, exhibit close connections with genes reported to be involved in potential immune mechanisms influencing LGG prognosis. Additionally, we have discovered that these eight significant genes have close connections with genes belonging to different signaling pathways that have been reported to be associated with the prognosis of GBM. Therefore, we believe these eight noteworthy identified genes are not limited to the Wnt/β-catenin signaling pathway but may also serve as potentially crucial key genes that interact with various signaling pathways important in glioma tumorigenesis. We have added the results of the additional bioinformatics network analysis as Figures  4 and 5 and incorporated related sentences into the Results and Discussion sections of the revised version of the manuscript, as follows.

Results

3.5. Functional gene ontology and pathway network analyses

Overall, the protein interaction network between the eight significant genes and the β-catenin gene shown in Fig. 3 can be summarized briefly as follows: the RPSA, RPL19, RPL18A, and RPS12 genes are closely correlated and are associated with peptide bond formation in the nucleus. FRAT1 is associated with the β-catenin phosphorylation cascade. RPS12 and FRAT1 are associated with positive regulation of the Wnt signaling pathway. BMP2 is associated with cellular differentiation and organ induction. BMP2 is associated with both positive and negative Wnt signaling. FSTL1 and CER1 are associated with blocking BMP signaling.

In this study, we observed that the genes associated with the prognosis of LGG patients seemed to be solely focused on the WNT/β-catenin signaling pathway. Considering that this may result in a biased perspective, we expanded our scope to investigate whether these eight noteworthy genes also interacted with genes from other signaling pathways known to be related to glioma prognosis. Fig. 4 shows a tightly woven interaction network between the RP family members (RPSA, RPL18A, RPL19, and RPS12) and genes potentially associated with the immune mechanisms of LGG, which have been reported to be related to the prognosis of patients with LGG (CD2, SPN, IL18, PTPRC, GZMA, and TLR7).[15]

Fig. 4. Bioinformatics network analysis visualizing the interactions between eight significant genes identified in this study (CER1, FRAT1, FSTL1, RPSA, BMP2, RPL18A, RPL19, and RPS12) and genes that are potentially involved in regulating the immune microenvironment and serve as independent prognostic markers for LGG (CD2, SPN, IL18, PTPRC, GZMA, and TLR7). The network was generated using Cytoscape with functional GO terms and biological pathways enrichment.

CER1, cerberus 1; FRAT1, FRAT regulator of WNT signaling pathway 1; FSTL1, follistatin-like 1; RPSA, ribosomal protein SA; BMP2, bone morphogenetic protein 2; RPL18A, ribosomal protein L18A; RPL19, ribosomal protein L19; RPS12, ribosomal protein S12; LGG, lower grade glioma; SPN, sialophorin; IL, interleukin; PTPRC; protein tyrosine phosphatase receptor type C; GZMA, granzyme A; TLR7, toll like receptor 7; GO, gene ontology.

These findings suggest that the RP family, traditionally associated with Wnt/β-catenin signaling, may have a broader role in influencing the immune landscape within LGG, potentially impacting patient prognosis. In addition, we recently identified 12 independent genes across 10 oncogenic signaling pathways significantly associated with mortality and disease progression in patients with GBM.[11] Therefore, we explored the potential connections between these 12 significant genes from the 10 oncogenic pathways and the eight noteworthy genes identified in this study. Fig. 5. presented a detailed network analysis revealing close interactions between the CER1, FRAT1, FSTL1, and BMP2 genes and 12 independent genes identified across 10 oncogenic signaling pathways that are significantly associated with prognosis in GBM patients.

Fig. 5. Bioinformatics network analysis visualizing the interactions between eight significant genes discovered in this research (CER1, FRAT1, FSTL1, RPSA, BMP2, RPL18A, RPL19, and RPS12), and twelve independent genes from ten oncogenic signaling pathways (E2F2 [cell cycle signaling pathway], CTBP2 [Notch signaling], MAFF [Nrf2 signaling], SLC2A3 [Nrf2 signaling], ECSIT [PI3K signaling], HSP90B1 [PI3K signaling], TNFRSF1A [PI3K signaling], PAK1 [RTK signaling], ID4 [TGF-β signaling], DDB2 [p53 signaling], MDM2 [p53 and cell cycle signaling], and DKK3 [Wnt/β-catenin signaling]) that have been significantly associated with prognosis in patients with GBM. The network was generated using Cytoscape with functional enrichment of GO terms and biological pathways.

CER1, cerberus 1; FRAT1, FRAT regulator of WNT signaling pathway 1; FSTL1, follistatin like 1; RPSA, ribosomal protein SA; BMP2, bone morphogenetic protein 2; RPL18A, ribosomal protein L18A; RPL19, ribosomal protein L19; RPS12, ribosomal protein S12; E2F2, E2F transcription factor 2; CTBP2, C-terminal-binding protein 2; MAFF, MAF bZIP transcription factor F; Nrf2, nuclear factor erythroid 2-related factor 2; SLC2A3, solute carrier family 2 member 3; ECSIT, evolutionarily conserved signaling intermediate in Toll pathways; PI3K, phosphatidylinositol 3-kinase; HSP90B1, heat shock protein 90 kDa beta member 1; TNFRSF1A, tumor necrosis factor receptor superfamily member 1A; PAK1, p21 activated kinase 1; RTK, receptor tyrosine kinase; ID4, inhibitor of DNA binding 4; TGF-β, transforming growth factor beta; DDB2, damage-specific DNA-binding protein 2; MDM2, mouse double minute 2 homolog; DKK3, dickkopf-3; GBM, glioblastoma multiforme; GO, gene ontology.

These findings indicate substantial crosstalk between Wnt/β-catenin signaling and various other oncogenic pathways, underscoring the role of Wnt/β-catenin signaling in the broader oncogenic landscape and its potential impact on glioma prognosis.

  1. Discussion

Among these eight significant genes, positive correlations were observed between CER1 and FSTL1 and between FRAT1, BMP2, and the RP family (RPSA, RPS12, RPL18A, and RPL19). Through bioinformatics network analysis, we found that the eight significant genes identified in our study, correlated with the prognosis of LGG and associated with the Wnt signaling pathway, exhibit close connections with genes reported to be involved in potential immune mechanisms influencing LGG prognosis. Additionally, we discovered that these eight significant genes had close connections with genes belonging to different signaling pathways that have been reported to be associated with the prognosis of GBM. Therefore, we believe these eight noteworthy identified genes are not limited to the Wnt/β-catenin signaling pathway but may also serve as potentially crucial key genes that interact with various signaling pathways important in glioma tumorigenesis.

Up to 70% of low-grade gliomas transform into high-grade gliomas within 10 years, and this transformation is associated with changes in several genes and molecular pathways.[19,20]

Reviewer 2 Report

Comments and Suggestions for Authors

The authors come up with a  bioinformatics capture of Significant genes associated with mortality and disease progression in grade II and III glioma and come up with the role played by the Wnt/β-catenin pathways in tumorigenesis.  They show how the tumorigenesis is initiated, as against antagonism associated with BMP9 and other genes.  They take publicly available databases in filtering out wide number of geneic data, find the overall survival and progression free survival and perform the downstream integrated systems bioinformatics approach using STRING, Cytoscape and aided by statistical interpretation.  The manuscript is generally written well with a good hypothesis, albeit  with a major limitation of no validation in vitro. However, works of such types need no validation but statistical concordance is to be checked which they did, IMHO.

What intrigues me is that the authors mention that they "report"  buy citing their previous study ( reference 8) from brain gene at last database.  I don't think they generated this data, and it is again a publicly available datasets on which they laid this hypothesis.  

The gene names/aliases/ must be italicised and in some tables, for example Table 3 they are not italicised

The STRING cutoff they sued was 0.4, assuming 40%  chance of considering those interacting partners.  However, there is no mention on whether any genes are coexpressed on what is the weightage given to physical interactions etc.

The Cytoscape was used to visualize and of course, perform some analyses which they could draw a conclusion on whether all OS/PFS genes were considered

The mRNA expression data of 515 patients with grade II and III glioma from the TCGA database were considered and a large numbers were female, is there any specific reason?

The four genes associated with significant mortality and disease progression in patients with grade II and III glioma could also be coexpress, colocalised. Is there any data available on this? Was such visualization done using STRING/Cytoscape 

In Table 3, again, some rows were in bold but a few oligoastrocytoma  data where p vale is 0.18 were also picked. Pl justify 

Likewise, Karnofsky Performance, in my understanding , greatre is the value, greater is the survival and this is negatively correlated with p value heuristics. Pl check. 

Pl keep conclusions as a separate head

Minor but essential:

L 122:  Pl remove "with"  after associated 

Author Response

Significant genes associated with mortality and disease progression in grade II and III glioma

Bo Mi Choi, Jin Hwan Cheong, Je Il Ryu, Yu Deok Won, Kyueng-Whan Min, Myung-Hoon Han

Responses to reviewer comments

Editorial Board

Biomedicines

Dear Reviewers and Editorial Staffs of Biomedicines:

We sincerely thank you for your thorough review of our manuscript. We have considered the comments and suggestions with care and addressed the critical issues highlighted by the reviewers. 

These responses and revisions have resulted from the authors’ hard work and sincerity in addressing the suggestions of the reviewers and editors. We are grateful for your guidance in improving the scientific and literary qualities of our work. During the revision process, we underwent English revisions throughout the entire paper.

This manuscript has not been published or presented elsewhere in part or in its entirety and is not under consideration by another journal. Given the use of anonymized data and the retrospective nature of the study, informed consent was waived. The study design was approved by the appropriate ethics review board. We have read and understood your journal’s policies and believe that neither the manuscript nor the study violates them. The authors declare no conflicts of interest.

Thank you for your consideration. I look forward to hearing from you.

Sincerely,

Myung-Hoon Han, M.D., Ph.D.

Department of Neurosurgery, Hanyang University Guri Hospital, 153 Gyeongchun-ro,

Guri, Gyonggi-do, Republic of Korea

Tel: +82-31-560-2326

Fax: +82.31-560-2327

  Kyueng-Whan Min, M.D., Ph.D.

Department of Pathology Uijeongbu Eulji Medical Center, Eulji University School of Medicine, Uijeongbu, Gyeonggi-do, Republic of Korea

Tel: +82-31-560-2496; Fax: +82-31-560-2339

ORCID ID: 0000-0002-4757-9211

Reviewer #2

What intrigues me is that the authors mention that they "report"  buy citing their previous study ( reference 8) from brain gene at last database.  I don't think they generated this data, and it is again a publicly available datasets on which they laid this hypothesis. 

Thank you for your comment. We have revised the sentence as follows.

  1. Materials and Methods

2.1. Study patients

We recently published a study using 525 glioblastoma multiforme (GBM) cases with reported glioma cases (619 glioblastoma multiforme [GBM] cases and 530 lower grade glioma [LGG] cases), using information on mRNA expression obtained from the TCGA database.[8]

The gene names/aliases/ must be italicised and in some tables, for example Table 3 they are not italicised.

Thank you for your comment. As per Reviewer 2’s suggestion, we have italicized all gene names in the revised version of the manuscript.

The STRING cutoff they sued was 0.4, assuming 40%  chance of considering those interacting partners.  However, there is no mention on whether any genes are coexpressed on what is the weightage given to physical interactions etc.

Thank you for your comment. As per the suggestion of Reviewer 2, we have added the following sentences to the results section of the revised version of the manuscript.

  1. Results

3.3. Genes related to the four selected significant genes associated with both OS and PFS in glioma patients

The results indicated that: (1) CER1 exhibited close connections with BMPR2 (interaction score, 0.921), BMP2 (interaction score, 0.930), BMP4 (interaction score, 0.946), ACVR1B (interaction score, 0.921), and TDGF1 (interaction score, 0.947); (2) FRAT1 demonstrated significant interactions with CTNNB1 (interaction score, 0.953), DVL1 (interaction score, 0.972), AXIN1 (interaction score, 0.996), GSK3B (interaction score, 0.990), and APC (interaction score, 0.928); (3) FSTL1 was closely connected to BMP2 (interaction score, 0.953), TLR2 (interaction score, 0.972), DIP2A (interaction score, 0.996), BMP4 (interaction score, 0.990), and SPARC (interaction score, 0.928); and (4) RPSA showed a very close connection with RPL19 (interaction score, 0.999), RPS12 (interaction score, 0.999), RPL18A (interaction score, 0.999), RPL35 (interaction score, 0.999), and RPS (interaction score, 0.999) (Fig. 2A). Eighteen genes were found to be closely related to the four selected genes. We also constructed a heat map to show the log2 fold changes in the expression of these 18 genes in patients with grade II and III gliomas (Fig. 2B).

The mRNA expression data of 515 patients with grade II and III glioma from the TCGA database were considered and a large numbers were female, is there any specific reason?

Thank you for your comment. GBM exhibits a higher prevalence in males, yet the sex ratio for lower-grade glioma, which is the subject of this paper, is known to be nearly equal. This is corroborated by previous research, as stated: “The most obvious difference associated with sex in GBM is its incidence: the incidence of GBM is 1.6 times higher in men than in women. Whereas the incidence of low-grade glioma is nearly similar in men and women (Carrano, A.; Juarez, J.J.; Incontri, D.; Ibarra, A.; Cazares, H.G. Sex-Specific Differences in Glioblastoma. Cells 2021, 10, 1783).” Additionally, another study utilizing the same TCGA LGG data as ours also reported an identical count of female patients, with 230 out of 515 (Zhu, W, et al. Cuprotosis Clusters Predict Prognosis and Immunotherapy Response in Low-Grade Glioma. Apoptosis 2024, 29, 169–190).

The Cytoscape was used to visualize and of course, perform some analyses which they could draw a conclusion on whether all OS/PFS genes were considered

The four genes associated with significant mortality and disease progression in patients with grade II and III glioma could also be coexpress, colocalised. Is there any data available on this? Was such visualization done using STRING/Cytoscape

Thank you for your valuable comment. Following the suggestion of Reviewer 2, we investigated the co-expression of four genes associated with significant mortality and disease progression in patients with grade II and III glioma through STRING, and the results were as follows.

Upon examining the co-expression among the four significant genes, it appears that the co-expression weightage within the RP family, including RPSA, is so strong that it suggests co-expression occurs exclusively within the RP family.

Therefore, we performed additional network analysis using Cytoscape to examine the interactions and subcellular localizations among the four significant genes: CER1, FRAT1, FSTL1, and RPSA. This analysis enabled us to observe intricate interactions among these genes, as well as to identify their subcellular localizations. The findings from this analysis have been incorporated into the revised version of the manuscript as Supplementary Figure 2, as follows.

Supplementary Fig. 2

Supplementary Fig. 2. Bioinformatics network analysis visualizes the interactions and subcellular localization between four significant genes identified in this study (CER1, FRAT1, FSTL1, and RPSA). The network was generated using Cytoscape with functionally enriched GO terms and biological pathways.

GO, gene ontology.

And we have added the following sentences to the Results section.

  1. Results

3.2. Genes of the Wnt/β-catenin pathway associated with OS and PFS in glioma patients

We additionally conducted network analysis to investigate the interactions and subcellular localizations among four significant genes (CER1, FRAT1, FSTL1, and RPSA) involved in the Wnt/β-catenin pathway (Supplementary Fig. 2). Our analysis revealed intricate interactions among these four significant genes, as well as the distribution of their subcellular localization. We also presented the Kaplan–Meier survival curves for the OS and PFS of the study patients based on the upper and lower median groups of expression of these four genes (Fig. 1D-G).

Additionally, we utilized Cytoscape to perform further network analysis aimed at examining the relationships between the eight significant genes involved in Wnt signaling that we identified and other genes that, while associated with different pathways, have been reported to relate to the prognosis of glioma. The results of this analysis have been incorporated into the revised version of the manuscript as follows.

Results

3.5. Functional gene ontology and pathway network analyses

Overall, the protein interaction network between the eight significant genes and the β-catenin gene shown in Fig. 3 can be summarized briefly as follows: the RPSA, RPL19, RPL18A, and RPS12 genes are closely correlated and are associated with peptide bond formation in the nucleus. FRAT1 is associated with the β-catenin phosphorylation cascade. RPS12 and FRAT1 are associated with positive regulation of the Wnt signaling pathway. BMP2 is associated with cellular differentiation and organ induction. BMP2 is associated with both positive and negative Wnt signaling. FSTL1 and CER1 are associated with blocking BMP signaling.

In this study, we observed that the genes associated with the prognosis of LGG patients seemed to be solely focused on the WNT/β-catenin signaling pathway. Considering that this may result in a biased perspective, we expanded our scope to investigate whether these eight noteworthy genes also interacted with genes from other signaling pathways known to be related to glioma prognosis. Fig. 4 shows a tightly woven interaction network between the RP family members (RPSA, RPL18A, RPL19, and RPS12) and genes potentially associated with the immune mechanisms of LGG, which have been reported to be related to the prognosis of patients with LGG (CD2, SPN, IL18, PTPRC, GZMA, and TLR7).[15]

Fig. 4. Bioinformatics network analysis visualizing the interactions between eight significant genes identified in this study (CER1, FRAT1, FSTL1, RPSA, BMP2, RPL18A, RPL19, and RPS12) and genes that are potentially involved in regulating the immune microenvironment and serve as independent prognostic markers for LGG (CD2, SPN, IL18, PTPRC, GZMA, and TLR7). The network was generated using Cytoscape with functional GO terms and biological pathways enrichment.

CER1, cerberus 1; FRAT1, FRAT regulator of WNT signaling pathway 1; FSTL1, follistatin-like 1; RPSA, ribosomal protein SA; BMP2, bone morphogenetic protein 2; RPL18A, ribosomal protein L18A; RPL19, ribosomal protein L19; RPS12, ribosomal protein S12; LGG, lower grade glioma; SPN, sialophorin; IL, interleukin; PTPRC; protein tyrosine phosphatase receptor type C; GZMA, granzyme A; TLR7, toll like receptor 7; GO, gene ontology.

These findings suggest that the RP family, traditionally associated with Wnt/β-catenin signaling, may have a broader role in influencing the immune landscape within LGG, potentially impacting patient prognosis. In addition, we recently identified 12 independent genes across 10 oncogenic signaling pathways significantly associated with mortality and disease progression in patients with GBM.[11] Therefore, we explored the potential connections between these 12 significant genes from the 10 oncogenic pathways and the eight noteworthy genes identified in this study. Fig. 5. presented a detailed network analysis revealing close interactions between the CER1, FRAT1, FSTL1, and BMP2 genes and 12 independent genes identified across 10 oncogenic signaling pathways that are significantly associated with prognosis in GBM patients.

Fig. 5. Bioinformatics network analysis visualizing the interactions between eight significant genes discovered in this research (CER1, FRAT1, FSTL1, RPSA, BMP2, RPL18A, RPL19, and RPS12), and twelve independent genes from ten oncogenic signaling pathways (E2F2 [cell cycle signaling pathway], CTBP2 [Notch signaling], MAFF [Nrf2 signaling], SLC2A3 [Nrf2 signaling], ECSIT [PI3K signaling], HSP90B1 [PI3K signaling], TNFRSF1A [PI3K signaling], PAK1 [RTK signaling], ID4 [TGF-β signaling], DDB2 [p53 signaling], MDM2 [p53 and cell cycle signaling], and DKK3 [Wnt/β-catenin signaling]) that have been significantly associated with prognosis in patients with GBM. The network was generated using Cytoscape with functional enrichment of GO terms and biological pathways.

CER1, cerberus 1; FRAT1, FRAT regulator of WNT signaling pathway 1; FSTL1, follistatin like 1; RPSA, ribosomal protein SA; BMP2, bone morphogenetic protein 2; RPL18A, ribosomal protein L18A; RPL19, ribosomal protein L19; RPS12, ribosomal protein S12; E2F2, E2F transcription factor 2; CTBP2, C-terminal-binding protein 2; MAFF, MAF bZIP transcription factor F; Nrf2, nuclear factor erythroid 2-related factor 2; SLC2A3, solute carrier family 2 member 3; ECSIT, evolutionarily conserved signaling intermediate in Toll pathways; PI3K, phosphatidylinositol 3-kinase; HSP90B1, heat shock protein 90 kDa beta member 1; TNFRSF1A, tumor necrosis factor receptor superfamily member 1A; PAK1, p21 activated kinase 1; RTK, receptor tyrosine kinase; ID4, inhibitor of DNA binding 4; TGF-β, transforming growth factor beta; DDB2, damage-specific DNA-binding protein 2; MDM2, mouse double minute 2 homolog; DKK3, dickkopf-3; GBM, glioblastoma multiforme; GO, gene ontology.

These findings indicate substantial crosstalk between Wnt/β-catenin signaling and various other oncogenic pathways, underscoring the role of Wnt/β-catenin signaling in the broader oncogenic landscape and its potential impact on glioma prognosis.

  1. Discussion

Among these eight significant genes, positive correlations were observed between CER1 and FSTL1 and between FRAT1, BMP2, and the RP family (RPSA, RPS12, RPL18A, and RPL19). Through bioinformatics network analysis, we found that the eight significant genes identified in our study, correlated with the prognosis of LGG and associated with the Wnt signaling pathway, exhibit close connections with genes reported to be involved in potential immune mechanisms influencing LGG prognosis. Additionally, we discovered that these eight significant genes had close connections with genes belonging to different signaling pathways that have been reported to be associated with the prognosis of GBM. Therefore, we believe these eight noteworthy identified genes are not limited to the Wnt/β-catenin signaling pathway but may also serve as potentially crucial key genes that interact with various signaling pathways important in glioma tumorigenesis.

Up to 70% of low-grade gliomas transform into high-grade gliomas within 10 years, and this transformation is associated with changes in several genes and molecular pathways.[19,20]

In Table 3, again, some rows were in bold but a few oligoastrocytoma  data where p vale is 0.18 were also picked. Pl justify

Thank you for your comment. With the assistance of Reviewer 2, we corrected an instance where a p value of 0.064 in oligodendroglioma was mistakenly highlighted in bold.

Regarding the oligoastrocytoma data that Reviewer 2 mentioned, the p value was not 0.18 but 0.018, which justifies our decision to keep it highlighted in bold.

Likewise, Karnofsky Performance, in my understanding , greatre is the value, greater is the survival and this is negatively correlated with p value heuristics. Pl check.

As evident from Table 1 or the Supplementary Data 1 we provided, the median Karnofsky Performance Score among the patients included in this study was 80. Consequently, as can also be observed in Table 3, the variation in Karnofsky Performance Score values among the patients is not significant enough to produce statistically significant results. This is reflected in the non-significant p values and hazard ratios of 0.99 or 1, which we believe to be of minimal academic relevance. Given that this study focuses on patients with lower-grade glioma, all of whom underwent surgery, it suggests that most patients enrolled in the study likely had a sufficiently good Karnofsky Performance Score to qualify for surgery at the time of enrollment.

Pl keep conclusions as a separate head

In response to your comment, we have ensured that the conclusions are now presented under a separate heading in the revised manuscript.

L 122:  Pl remove "with"  after associated

We have removed “with” following “associated” at line 122 in the revised manuscript.

Round 2

Reviewer 1 Report

Comments and Suggestions for Authors

The author has address the questions and suggestions given in the previous report. I have no further comments.

Reviewer 2 Report

Comments and Suggestions for Authors

I am satisfied with the changes rendered

Comments on the Quality of English Language

I am satisfied with the changes rendered

Language checks may be thoroughly checked